# Clinical Profile and Aspects of Differential Diagnosis in Patients with ME/CFS from Latvia

**DOI:** 10.3390/medicina57090958

**Published:** 2021-09-11

**Authors:** Angelika Krumina, Katrine Vecvagare, Simons Svirskis, Sabine Gravelsina, Zaiga Nora-Krukle, Sandra Gintere, Modra Murovska

**Affiliations:** 1Department of Infectology, Rīga Stradiņš University, 16 Dzirciema St., LV-1007 Riga, Latvia; 2Institute of Microbiology and Virology, Rīga Stradiņš University, 5 Ratsupites St., LV-1067 Riga, Latvia; katrine.vecvagare@rsu.lv (K.V.); simons.svirskis@rsu.lv (S.S.); sabine.gravelsina@rsu.lv (S.G.); zaiga.nora@rsu.lv (Z.N.-K.); modra.murovska@rsu.lv (M.M.); 3Department of Family Medicine, Rīga Stradiņš University, 16 Dzirciema St., LV-1007 Riga, Latvia; sandra.gintere@rsu.lv

**Keywords:** myalgic encephalomyelitis/chronic fatigue syndrome, symptoms, diagnosis, visual analogue scale

## Abstract

*Background and objectives*: There is still an uncertainty regarding the clinical symptomatology and the diagnostic criteria in terms of myalgic encephalomyelitis/chronic fatigue syndrome (ME/CFS), as different diagnostic criteria exist. Our aim is to identify the core symptoms of ME/CFS in the outpatient setting in Riga; to distinguish symptoms in patients with ME/CFS and those with symptoms of fatigue; and to investigate patient thoughts on the onset, symptoms, treatment and effect of ME/CFS. *Materials and methods*: Total of 65 Caucasian patients from an ambulatory care setting were included in the study. Questionnaires, specialist evaluation of the patients and visual analogue scale (VAS) measurements were used to objectify the findings. *Results*: The study showed that ME/CFS with comorbidities is associated with a more severe disease. A negative correlation was found regarding an increase in age and number of current symptoms, as well as an increase in VAS score and the duration of fatigue and age in the ME/CFS without comorbidities group. *Conclusions*: Comorbidities tend to present with a more severe course of ME/CFS. Fatigue, myalgia, arthralgia and sleep disturbances tend to be more prevalent in the ME/CFS patients compared to the non-ME/CFS patients. VAS score has a tendency to decrease with age and duration of fatigue. Nonsteroidal anti-inflammatory drugs are the most commonly used pharmacological drug class that reduces ME/CFS symptoms.

## 1. Introduction

Myalgic encephalomyelitis (ME) or chronic fatigue syndrome (CFS) is post-viral or post-infectious fatigue syndrome or systemic exertional intolerance disease (SEID) that affects the functioning ability of a person and reduces the energy below the level that is considered the average. It is a complex and multifactorial disease that not only dysregulates the central nervous system, immune system and cellular energy metabolism, but also influences physical and cognitive state [1].

Nowadays there are several terms that are being used in the literature to describe ME/CFS. Historically CSF and ME were used separately, as different nosologic entities, but when Federal Health Agencies in the United States of America combined them together in 2016, ME/CFS has been used as an umbrella term to identify multi-systemic, chronic disease that causes physical, cognitive, or emotional exertion [2]. SEID is a relatively new term that has been proposed by the Institute of Medicine (IOM) in 2015 [3] and introduced based on the characteristic, central elements of the disease. No matter which diagnostic criteria are being used, the recent publications aim to declare that post-exertional malaise (PEM) is one of the key symptoms [4]. In this report the term ME/CFS will be used.

As to the statistics of ME/CFS, the numbers vary and depend on the country and research. The prevalence is from 0.42% to 2.54% worldwide [5] and there are from one million to over five million people suffering from ME/CFS in Europe [4].

Several aetiological scenarios are discussed in terms of ME/CFS, but it is still considered multifactorial spectrum of illness with controversial, complex and unknown aetiology that is triggered by different factors and happens to develop in people with predisposition. There have been investigations in terms of neurological, immunological, endocrine, genetic and infectious causes, but none of these are considered the leading one [6].

As regards the diagnostics of ME/CFS, there is no single golden standard that is accepted worldwide, but several criteria systems have been used depending on the country or healthcare centre. In general, the diagnosis is based on the patient’s subjective symptoms and differential diagnostics to exclude other pathologies, because there are no biomarkers or other tests that could serve to objectify this process.

In the last 30 to 40 years approximately 20 different diagnostic criteria systems have been proposed. One of the most commonly used is the Fukuda criteria (FC) [7], more recently the Canadian Consensus Criteria (CCC) [8] have been proposed, as well as the International Consensus Criteria (ICC) [1], the Oxford criteria (OC) [9] and the criteria released by the IOM in 2015 [10], the latter having received international recognition [4]. FC (1994) commonly serves as a diagnostic tool in research purposes. As to the recent suggestion from the European Network on Myalgic Encephalomyelitis/Chronic Fatigue Syndrome (EUROMENE) expert consensus, PEM should also be included in the core symptoms of the Fukuda criteria, to decrease the risk of hyperdiagnostics [4].

To better asses ME/CFS symptoms and to objectify them, several questionnaires and functional tests are being used: “UK ME/CFS Participant Questionnaire” [11]; “DePaul Symptom Questionnaire” (DSQ) [5,12]; “The RAND-36 Item Health Survey” [13,14]. There are certain strengths for each of the questionnaires, but they all have been used in both clinical and research purposes [15]. To evaluate the functioning ability, the most commonly used scales are: “Work and Social Adjustment Scale (WSAS) [16,17], “Energy Index Point Score” [18], “The Lawton Instrument Activities of Daily Living Scale” [19], VAS [20], “SF-36” [21,22], EQ-5D [23,24] and others. To evaluate sleep disturbances: “Sleep Assessment Questionnaire” [25], “Pittsburg Sleep Quality Index” [26], “PROMIS Sleep questionnaire” [27], as well as The Epworth Sleepiness Scale to detect daytime sleepiness [4] are being used.

To decrease the risk of hyperdiagnostics, several differential diagnosis should be excluded. Besides that, there are certain diseases that usually manifest together with ME/CFS and do not rule out the diagnosis of ME/CFS. Some of the overlap syndromes are: allergies, fibromyalgia, irritable bowel syndrome, postural orthostatic tachycardia syndrome, hypotension, hypogonadism and premature menopause, sleep disorders, hypersensitivities, hypoglycaemia, mitral valve prolapse, metabolic syndrome, vitamin B12 deficiency, endometriosis and others [28].

Although ME/CFS is a disabling disease with an impact on functional status and quality of life, no specific treatment or cure for ME/CFS exist, it tends to be individualised and usually vary from case to case. Nevertheless, both - pharmacological and non-pharmacological methods, as well as alternative medicine are used to reduce the symptoms and improve the quality of life and well-being [12]. But it should be noted that results of the research have been controversial, leading to reduction of the symptoms, aggravation of the symptoms, being ineffective or causing side effects. That is why these measurements should be done under control and the choice of treatment should be based according to the national guidelines.

To conclude, it is clearly seen that ME/CFS remains a challenge for medical specialists. As this disease has unclear aetiology, symptomatic variability and there are no common grounds for unified diagnostic criteria, it is challenging to find the best treatment option. However, a wide range of research is being conducted in pharmacological, non-pharmacological, as well as alternative medicine fields, therefore new strategies and potential improvements are still to come to improve the work of the clinicians and to raise the quality of life of the patients.

## 2. Materials and Methods

### 2.1. Patient Selection and Eligibility Criteria

This prospective observational study includes a Latvian population of 65 Caucasian participants (43 females, 22 males) undergoing outpatient treatment in Rīga Stradiņš University ambulance in Riga, Latvia from April 2020 to May 2021. Age ranged from 23–78 years in females and 21–72 years in males. The average age ± SD for both genders was 47.4 ± 14.92 years (47.40 ± 14.66 in females and 47.41 ± 15.78 in males).

The inclusion criteria were as follows:18 years or older;Patient or legally authorised representative capable to give informed consent;Fatigue lasting for at least six consecutive months;Subjective symptoms of fatigue for more than six months or previously diagnosed with ME/CFS using the Fukuda et.al diagnostic criteria;Meets the neurologic criteria;Fatigue includes PEM as a compulsory symptom.

The exclusion criteria were as follows:Younger than 18 years;Pregnancy or breast feeding;Inability to obtain or declined informed consent;Cancer, radiation, chemotherapy at the time of enrolment;Acute infectious or inflammatory diseases;Previously diagnosed depression and/or any other psychiatric disorder;Substance abuse and/or eating disorder within two years of the onset of ME/CFS symptoms;Obesity with body mass index greater than 45;Primary brain disorder.

Patient selection was made by a qualified physician (infectologist, neurologist or general practitioner), specialised in ME/CFS diagnostics, who determined patient’s suitability depending on one’s clinical expertise. The selection was based on the new-onset fatigue symptoms, previously reported fatigue symptoms registered in medical histories, as well as a previous diagnosis of ME/CFS. All patients were observed by the physician, who reported demographic, medical, occupational and additional information.

### 2.2. Symptom Registration

Data were collected as part of care at an ambulatory outpatient health care facility. First, the participants were informed about the research, its purposes, their participation and then an informed consent was signed. Second, if the patients agreed, we asked to fill in questionnaires in the waiting room by hand. Of the 65 patients all 65 individuals returned the questionnaire. After completion, the patients were asked to share their questions and comments with a certified specialist, they were consulted and a VAS score was measured.

All patients were interviewed with questionnaires to evaluate various categories. To examine the symptom pattern in ME/CFS patients, we used adapted semi-structured interview questions created by Minnock et.al [29]. The questions were structured in six sections: causes and triggers of fatigue; character of fatigue; current symptoms; comorbidities; solutions for fatigue; and its influence on work disability. Multiple choice answers were provided for each question.

Regarding sleep disturbances, we included a self-reported questionnaire—Athens Insomnia Scale 8 [30]—to assess insomnia symptoms, which included the evaluation in various sleep-related questions: sleep induction, awakenings during the night, final awakening, sleep quality, well-being during the day, functioning capacity during the day and sleepiness during the day. We used the cut-off value of ≥ six points for the confirmation of sleep disturbances.

VAS, ranging from zero to ten was also measured for all patients to assess the disease-related pain intensity.

To better evaluate the differences in terms of symptoms, first, we divided the respondents into three groups—patients without ME/CFS presenting with symptoms of fatigue (*n =* 10), patients diagnosed with ME/CFS according to the Fukuda et.al criteria (*n* = 19) and patients diagnosed with ME/CFS according to the Fukuda et.al criteria, who have at least one comorbidity, which might be affecting the symptom severity and pattern of fatigue (*n* = 36). In some situations, we combined the two groups with the diagnosis of ME/CFS (*n* = 55) to better emphasise the differences between ME/CFS and non-ME/CFS patients. Second, based on the patient’s self-reported answers to the questionnaires, the answers were graded by our specialists according to the severity and a total score calculated, so that the correlation analysis and comparison regarding different patterns of fatigue could be made.

### 2.3. Statystical Analysis

Descriptive and advanced statistical analysis, as well as graphing were done using GraphPad Prism V.9.1 for macOS (GraphPad Software, Inc., San Diego, CA, USA). The normality of the distribution of the studied data was checked by D’Agostino and Pearson, Anderson–Darling and Shapiro–Wilk normality tests. The homogeneity of variances was tested using F-test or Brown–Forsythe and Bartlett’s tests. To determine and assess the correlative associations between indicators of fatigue in predefined groups, the Spearman’s rank correlation test was performed. Between-group comparisons of summarised fatigue scores expressed in percentage were done by unpaired t-test or Brown-Forsythe and Welch ANOVA tests with Dunnett’s T3 multiple comparison test as post-hoc procedure.

As characteristic of central tendency, arithmetic mean with ± standard deviation (SD) was applied. A *p* value < 0.05 was considered statistically significant for all tests.

### 2.4. Ethical Consideration

All of the participants received the information regarding research ethical considerations, description of the research, including the aim, the design and potential results of the inquiry, as well as an informed consent prior to study inclusion. Confidentiality was guaranteed and research subjects were informed about withdrawing from participation without any consequences.

## 3. Results

### 3.1. Subject Characteristics in ME/CFS and Non/ME/CFS Patients

Overall, there were 55 patients diagnosed with ME/CFS—with or without comorbidities. As regards results in these two groups, most of the respondents (58%) had been having fatigue for the last year or last two years (29%), whereas the minority—for the last six months (10%). In comparison, patients not diagnosed with ME/CFS reported having fatigue for the last year (60%) or the last six months (40%).

Asked about the onset of fatigue, most patients in the ME/CFS groups considered that emotional (24%) or physical (22%) stress is a contributing factor, whereas 16% reported that it developed gradually with a progression of an underlying chronic disease and 15%—because of sleep disturbances. In the group without ME/CFS on the other hand, most of the patients (60%) reported that they could not remember or identify the onset or the reason for fatigue and none reported that it begun together with a progression of a chronic disease. Emotional stress (30%) was considered a cause for fatigue in more cases than physical stress (15%) in the non-ME/CFS group.

Most patients in both groups with ME/CFS (65%), as well as patients in the non-ME/CFS (50%) group stated that fatigue is constant and invariable throughout the day, whereas for 16% of respondents in both of the ME/CFS groups, compared to none in the non-ME/CFS group fatigue was more severe in the mornings.

Regarding the core symptoms, apart from fatigue with PEM (100%), myalgia (96%), headache (87%), arthralgia (86%) and difficulty concentrating (84%) are the five most common ones in the two groups with ME/CFS, whereas in the non-ME/CFS group those are: headache (91%), myalgia (73%), difficulty concentrating (64%), neck stiffness (64%) and fatigue (55%). All of the symptoms are listed in the Table 1. A graphical representation of the differences regarding symptoms is shown in the Figure 1. It shows that fatigue, myalgia, arthralgia and sleep disturbances are the main symptoms, which have a tendency to differ in ME/CFS patients compared to non-ME/CFS patients.

In most cases (93%) our respondents in both of the ME/CFS groups could not identify any first-degree relatives having similar symptoms of fatigue, but if such a tendency was reported (7%), then in all of the cases the relative was mother.

Considering the effect fatigue has on the employment status, almost all respondents (82%) in both—the ME/CFS and ME/CFS with comorbidities groups had reduced their workload or become unemployed.

Regarding comorbidities presenting together with ME/CFS, the overall prevalence was 66%. Fibromyalgia, chronic hepatitis and Lyme disease occurred in 20%, 9% and 5%, respectively. EBV, enterovirus infection each occurred in 5% of cases, whereas lymphadenopathy and anaemia were registered in 4% of cases. The schematic representation of all diagnosis can be assessed in the Figure 2, where comparison of two diagnostic groups can be seen—the group with ME/CFS (*n* = 19) and the one with ME/CFS and at least one comorbidity (*n* = 36). It must be noted that all infectious or inflammatory diseases were not in their acute phase at the time of the research.

Comparing self-reported treatment methods to decrease the symptoms of fatigue, in the non-ME/CFS group almost none of the respondents (90%) had found any solutions to decrease fatigue, whereas in both groups with ME/CFS 38% reported using help-self strategies, including physical activities, sleep hygiene, physiotherapy and walking, 38% had not found any solutions and 24% reported using pharmacological drugs, the most commonly used being non-steroidal anti-inflammatory drugs (90%).

The VAS score was also calculated for each individual and the average result was seven in both—the ME/CFS and ME/CFS with comorbidities group compared to six in the non-ME/CFS group.

### 3.2. Characteristic Differences in the Three, Previously Defined Groups

As shown in the Figure 3 there is a mild but significant increase of the overall scores of the pattern of fatigue, showing the lowest mean score in the patient group with non-ME/CFS diagnosis in comparison to ME/CFS (Figure 3a), however the level of the highest individual scores was established among those patients with ME/CFS with at least one comorbidity, indicating that comorbidities might be associated with a more severe course of the disease (Figure 3b).

Comparing the correlation coefficients in all three groups (Figure 4), those respondents in the non-ME/CFS group, who tend to identify more causes for fatigue and whose duration of fatigue was longer show an increased number of current symptoms (r = 0.59, r = 0.30, respectively). Additionally, more symptoms were identified in older patients (r = 0.30) and females (r = −0.48). In the ME/CFS group without comorbidities, on the other hand, more symptoms were identified in younger patients (r = −0.43) and in those who tend to mention less possible causes for their fatigue (r = −0.30). In both—non-ME/CFS and ME/CFS group without comorbidities more consequences of fatigue were identified by men than women (r = 30, r = 32, respectively).

Regarding VAS, there was a tendency for the VAS to be higher in younger patients (r = −0.55) and a negative correlation was found regarding an increase in VAS and the duration of fatigue in the ME/CFS group without comorbidities (r = −0.31). No correlations regarding VAS and previously mentioned parameters were found in the other two groups.

Only one correlation was found between the ME/CFS and the comorbidity group, showing that there is a tendency for the duration of fatigue to increase as the age increases (r = 0.40), as well as with an increasing age people tend to identify more possible causes for their fatigue (r = 0.32).

As to the correlation analysis in all of the groups together, only one correlation was identified, showing a tendency for the age to increase in the first (non-ME/CFS), the second (ME/CFS) and the third (ME/CFS + comorbidities) groups, respectively (r = 0.34).

## 4. Discussions

Comparably to the literature, where ME/CFS is said to be more commonly seen in women than men [31,32], in our study the tendency was similar (67% women, 33% men). The average age in our study was 49 years in persons diagnosed with ME/CFS (without and with comorbidities), which is more than stated in the literature, where the average age of onset is considered to be approximately 33 years [10].

According to the literature, in many cases ME/CFS follows a period of an acute infection [33,34,35,36], emotionally stressful incidents [37,38], or physical stressors. In one study the most common insidious event was considered an infection (64%) and stressful incidents (39%) [39], whereas our patients diagnosed with ME/CFS subjectively identified emotional (24%) or physical (22%) stress and to a lesser extent chronic diseases (16%)—to be possible triggers for their symptoms.

As regards the core symptoms of ME/CFS, there has been an ongoing discussion whether the diagnostic criteria identify the most prevalent symptoms and which criteria system would be the most suitable one. In our study the most prevalent symptoms apart from fatigue in the ME/CFS group were myalgia, headache, arthralgia, difficulty concentrating and neck stiffness. However, the symptoms that might be helpful in differentiating between ME/CFS and non-ME/CFS patients were fatigue, myalgia, arthralgia and sleep disturbances, which in our study were more commonly found in ME/CFS patients. Sleep disturbances were a common finding in our study (53%), which is less than in other research, where it has been found as common as in 79% of the subjects [40] showing a positive correlation regarding sleep problems and symptoms of ME/CFS [41]. As neurological and psychiatric comorbidities were excluded in our study, the identification and treatment of sleep disturbances might suggest a decrease in symptom severity. As it is stated in the literature [42], treatment of comorbidities might give promising results in attenuating the symptom severity. Although we did not make any follow-up of symptoms in this study, that would be a subject of interest in the future to evaluate the symptoms and investigate whether it is the most persistent symptom at follow-ups as stated in the literature [40] and whether treating these sleep disturbances could make any change in the current symptoms.

Some researchers have investigated the changes in the circadian rhythm in patients with fibromyalgia [43] and ME/CFS [44], showing that bright light during mornings has a tendency to improve function and pain sensitivity in fibromyalgia patients but has no effect in ME/CFS patients. In our study we concluded that ME/CFS patients report having fatigue as a rather steady symptom throughout the day (65%), but if there was a fluctuation in the severity during the day, then fatigue is more prevalent in the morning (16%), which corresponds to the information in other publications [45].

Our findings suggest that there is a rather limited association of fatigue and a positive family history. In most cases our respondents could not report any similar symptoms in their first-degree relatives in contrary to the other studies, where a positive family history showed a contribution to the predisposition of ME/CFS [46,47,48].

Whether the duration of the disease affects the outcome is still a debate in the literature. Some state that the duration of the illness might rather increase the ability to cope with the symptoms, in that way leading to less symptom prevalence [49], although others have found that those who have had ME/CFS for a longer period of time tend to have a more severe pattern of the disease [50,51]. Still other authors present with a finding that the duration of the illness does not predict the outcome [52]. In our study we found that there is a negative correlation regarding the age and current symptoms identified by the patients in the ME/CFS group without comorbidities, indicating that more symptoms are identified by younger patients. This caused us to think that younger patients might not yet have identified the coping mechanisms that help minimizing their symptoms. As to the effect of fatigue on work status in our study, 53% have reduced their workload or become professionally disabled, comparing to 65.1% [53] or 47% [39] in other latest studies and 40% presented in a systematic review of studies published from 1988 and 2001 [54]. In the current study those who were having the disease for 2 years or longer were more prone to change their workload and/or become professionally disabled, although follow-up surveillance would be needed to observe this tendency. Our findings regarding the effect of ME/CFS on the employment status are comparable to a study by Tiersky et al. [55], where the authors concluded that in most of the cases the patients are functionally affected and unemployed not only at the time of the diagnosis, but also on follow-ups. In our study 82% had reduced their workload or become unemployed in both of the ME/CFS groups. The authors of this study agree to the fact presented in previous research that the symptom severity decreases over time [4,39] as patients might be able to better manage their illness. It is substantiated by the fact that there was a negative correlation between the duration of fatigue, as well as age and an increase in VAS score in the ME/CFS group without comorbidities, showing no correlation in the non-ME/CFS group.

Although no treatment is found for ME/CFS, there are various strategies the patients use to decrease the severity of the symptoms. Interestingly, patients with ME/CFS were more prone to find a solution for their symptoms comparing to the non-ME/CFS group. According to the literature, the best way to decrease the symptom severity is to treat pain and sleep problems, because they might also be leading to a more severe pattern of other symptoms [4]. As it is seen in this study, nonsteroidal anti-inflammatory drugs are one of the most used pharmacological drugs and are considered effective in 90% of users in both of the ME/CFS groups. None of the patients reported taking supplements or undergoing cognitive behavioural therapy, which might be due to the fact that patients and their caregivers might not be informed about variable strategies to manage the disease.

## 5. Conclusions

This small-scale study provides important information on the evidence of symptom burden of ME/CFS patients from Riga, Latvia. ME/CFS with at least one comorbidity is associated with a more severe course of the disease. Fatigue, myalgia, arthralgia and sleep disturbances are the symptoms that have a tendency to be more prevalent in the ME/CFS compared to the non-ME/CFS patients. Symptoms in the ME/CFS group without comorbidities tend to decrease by increasing age, as well as more consequences of fatigue are identified by males in both—the non-ME/CFS group and the ME/CFS group without comorbidities. Younger patients and those who present with a shorter duration of the disease tend to have a higher VAS score in the ME/CFS group without comorbidities. An increase in age positively correlates with the duration of the disease, as well as potential causes identified in the ME/CFS group with at least one comorbidity. As to the treatment, the most frequently used pharmacological drug class that reduces the symptoms in patients with ME/CFS are nonsteroidal anti-inflammatory drugs.

It must be acknowledged that this paper can indicate the common patterns patients in particular region present with, although more research is needed to give access to a larger sample size and wider range of examples in order to better distinguish between ME/CFS and patients with fatigue symptoms (non-ME/CFS) in the clinical setting.

## Figures and Tables

**Figure 1 medicina-57-00958-f001:**
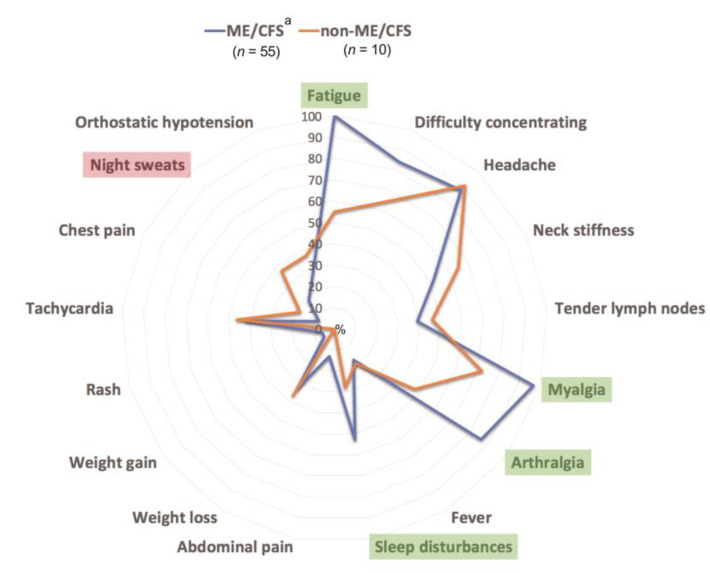
Comparative core symptom proportion in ME/CFS and non-ME/CFS patients using radial diagram; ^a^—all ME/CFS patients including those with comorbidities; ME/CFS: myalgic encephalomyelitis/chronic fatigue.

**Figure 2 medicina-57-00958-f002:**
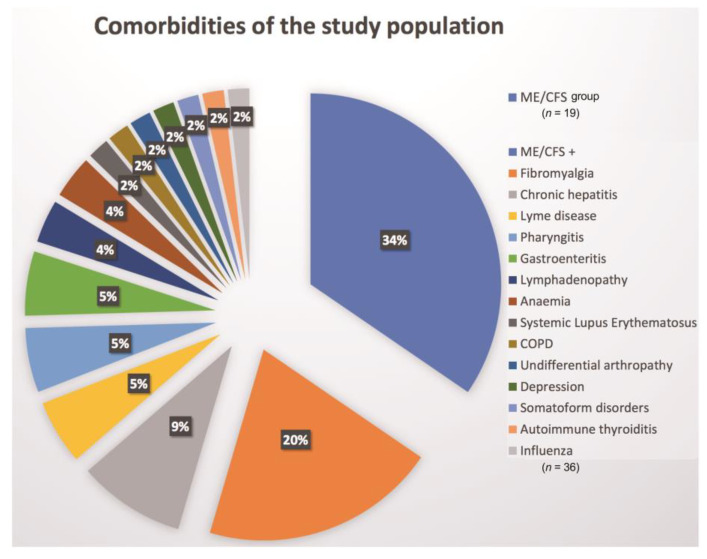
Representation of the proportion of ME/CFS (*n* = 19) and ME/CFS + comorbidities (*n* = 36) in the study population; COPD: Chronic obstructive pulmonary disease.

**Figure 3 medicina-57-00958-f003:**
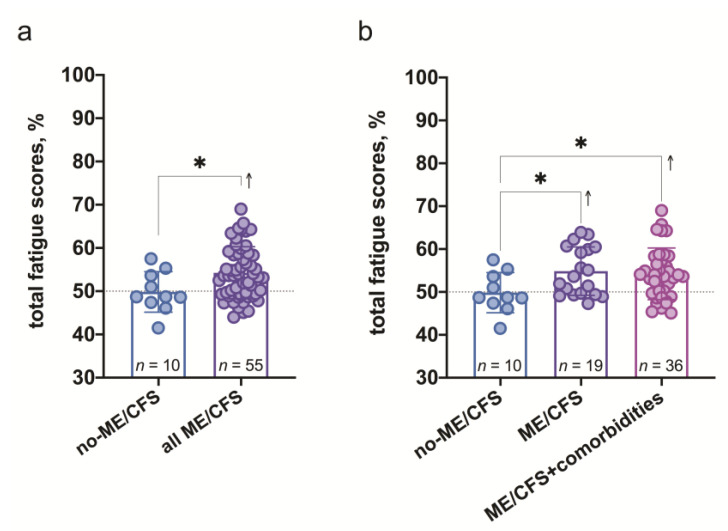
The overall scores of characteristics of fatigue expressed in %: (**a**)—in non-ME/CFS and all ME/CFS patients; (**b**)—in non-ME/CFS, ME/CFS and ME/CFS with at least one comorbidity patients. *—significance level *p* < 0.05 (**a**)—Unpaired *t*-test, (**b**)—Brown-Forsythe and Welch ANOVA tests with Dunnett’s T3 multiple comparison test as post-hoc procedure.

**Figure 4 medicina-57-00958-f004:**
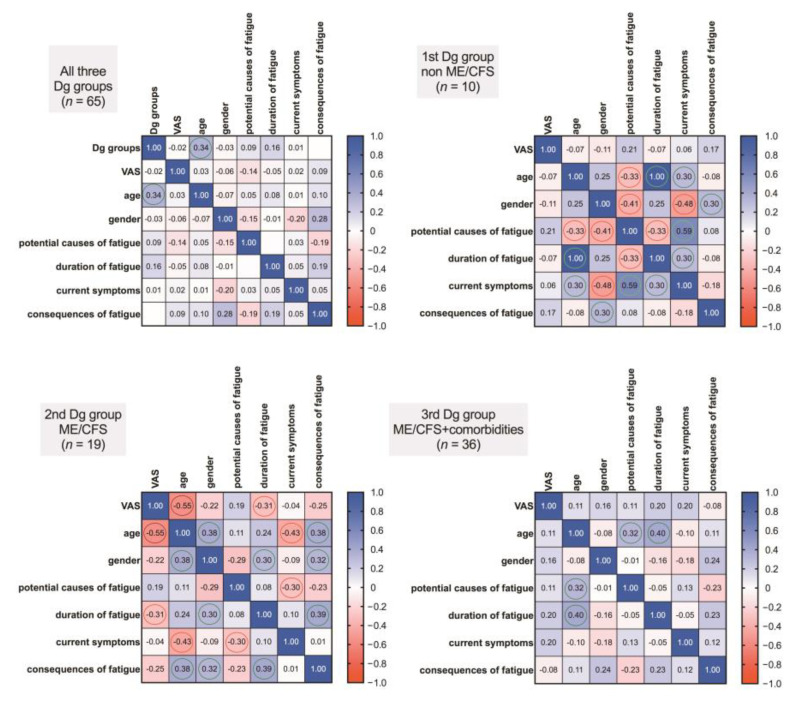
Presented correlograms showing the covariance of the studied variables in all three diagnostic groups—non-ME/CFS, ME/CFS and ME/CFS+comorbidities. The values in the squares represent Spearman’s rank correlation coefficients, showing the strength and direction of associations and the more pronounced ones are indicated by coloured circles (red—negative association, green—positive).

**Table 1 medicina-57-00958-t001:** Clinical signs of ME/CFS, ME/CFS with comorbidities and non-ME/CFS patients.

Clinical Features	ME/CFS, ME/CFS + Comorbidities Patients (*n* = 55)	95% CI
Fatigue	55 (100.0)	90.5–100.0
Myalgia	53 (96.4)	87.0–100.0
Headache	48 (87.3)	78.4–96.1
Arthralgia	47 (85.5)	76.7–94.2
Difficulty concentrating	46 (83.6)	74.9–92.3
Neck stiffness	29 (52.7)	45.8–59.7
Sleep disturbances	29 (52.7)	45.8–59.7
Tachycardia	23 (41.8)	35.7–48.0
Tender lymph nodes	21 (38.2)	32.3–44.1
Night sweats	19 (32.7)	27.3–38.2
Weight loss	18 (27.3)	22.3–32.2
Orthostatic hypotension	15 (18.2)	14.1–22.2
Fever	9 (16.4)	12.5–20.2
Abdominal pain	7 (12.7)	9.3–16.1
Chest pain	4 (7.3)	4.7–9.8
Weight gain	3 (5.5)	3.2–7.7
Rash	3 (5.5)	3.2–7.7
Clinical Features	Non-ME/CFS Patients (*n* = 10)	95% CI
Headache	10 (90.9)	81.9–100.0
Myalgia	8 (72.7)	64.6–80.8
Difficulty concentrating	7 (63.6)	56.1–71.2
Neck stiffness	7 (63.6)	56.1–71.3
Fatigue	6 (54.5)	47.5–61.5
Tender lymph nodes	5 (45.5)	39.0–51.9
Arthralgia	5 (45.5)	39.0–51.9
Tachycardia	5 (45.5)	39.0–51.9
Weight loss	4 (36.4)	30.6–42.1
Night sweats	4 (36.4)	30.6–42.1
Orthostatic hypotension	4 (36.4)	30.6–42.1
Sleep disturbances	3 (27.3)	22.3–32.2
Fever	2 (18.2)	14.1–22.2
Chest pain	2 (18.2)	14.1–22.2
Abdominal pain	0 (0.0)	0.0–0.0
Weight gain	0 (0.0)	0.0–0.0
Rash	0 (0.0)	0.0–0.0

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
