# Peer review of "Clinical Profile and Aspects of Differential Diagnosis in Patients with ME/CFS from Latvia"

_medicina, 2021, doi:10.3390/medicina57090958_

Round 1

Reviewer 1 Report

The paper is agreement with the title.

The paper is moderately relevant.

The topic is moderately original.

The paper adds a few new finding to other published material.

The paper is well written, the presentation is good, though somewhat overloaded.

The conclusions are consistent with the data presented.

There is no main question, since the paper is descriptive. 

This paper contains interesting information, is well-written and the results are presented adequately. However the most recent reference dates from 2019, whereas several papers on the same subject have been published since then. e.g. doi: 10.1016/j.medhy.2019.109444. The authors should update their literature search and add relevant references.

Author Response

Thank you very much for your thoughts and review on the manuscript. We accept and have fully taken them into account! As regards your remark about references and their publishing years we have provided a new reference dating from 2020 that could serve both as a good and fullfilling source of interest and show the actuality of the topic in general as well. 

All the changes we have made to the text are seen in the comments section in the document uploaded below. 

Reviewer 2 Report

The manuscript, Clinical profile and aspects of differential diagnosis in patients with ME/CFS from Latvia by Krumina et al. addresses the uncertainty in the existing diagnostic criteria of this disease as well as looks as comorbidities. Overall the paper is sound and the analytical methods appear appropriate, however, the use of parametric methods vs non-parametric methods should be justified by the acknowledgment of a normality test. I assume this was done but it should at least be stated. Also, there are some minor grammatical and technical writing issues that should be addressed. One example is that counting numbers below 10 should be spelled out, assuming there are no associated units. Also, it's not necessary to hyphenate words like comorbidity.   I really like the way the comorbidities were displayed. However, these are very minor issues, and generally speaking, the paper is very well written.

Author Response

Thank you so much for your constructive comments, which were very useful and helpful to improve the manuscript. We have taken them fully into account and you can find our comments on the changes made below:

  • Thank you for noticing this nuance. We have added the information regarding the normality of the distribution in the revised manuscript;
  • As to the counting numbers below 10, we accept and thank you for this remark. We have provided the document with counting numbers from 0 to 10 instead of numbers;
  • Regarding technical issues, we have corrected the spelling of the word "comorbidities". And we have inserted a gap between mathematical symbols throughout the text and images as well.

All of the changes can be found in the attached document, where they are outlined in the "comments" section. 
